# Immunometabolic Profile Associated with Progressive Damage of the Intestinal Mucosa in Adults Screened for Colorectal Cancer: Association with Diet

**DOI:** 10.3390/ijms242216451

**Published:** 2023-11-17

**Authors:** Celestino González, Sergio Ruiz-Saavedra, María Gómez-Martín, Aida Zapico, Patricia López-Suarez, Ana Suárez, Adolfo Suárez González, Carmen González del Rey, Elena Díaz, Ana Alonso, Clara G. de los Reyes-Gavilán, Sonia González

**Affiliations:** 1Department of Functional Biology, University of Oviedo, 33006 Oviedo, Spain; tinog@uniovi.es (C.G.); aiida.zapiico@gmail.com (A.Z.); lopezpatricia@uniovi.es (P.L.-S.); anasua@uniovi.es (A.S.); elenadr@uniovi.es (E.D.); alonsoana@uniovi.es (A.A.); 2Department of Microbiology and Biochemistry of Dairy Products, Instituto de Productos Lácteos de Asturias (IPLA-CSIC), 33300 Villaviciosa, Spain; sergio.ruiz@ipla.csic.es; 3Diet, Microbiota and Health Group, Instituto de Investigación Sanitaria del Principado de Asturias (ISPA), 33011 Oviedo, Spain; 4Digestive Service, Central University Hospital of Asturias (HUCA), 33011 Oviedo, Spain; 5Anatomical Pathology Service, Central University Hospital of Asturias (HUCA), 33011 Oviedo, Spain; carmenchugonzalezdelrey@gmail.com

**Keywords:** immunological factors, adipokines, cytokines, chemokines, colorectal cancer, intestinal mucosa, diet, bioactive compounds, xenobiotics

## Abstract

Environmental factors such as diet and lifestyle have been shown to influence the development of some intestinal mucosal lesions that may be precursors of colorectal cancer (CRC). The presence of these alterations seems to be associated with misbalanced immunological parameter levels. However, it is still unclear as to which immunological parameters are altered in each phase of CRC development. In this work, we aimed to study the potential relationships of immunological and metabolic parameters with diet in a CRC-related lesion context. Dietary information was obtained using an annual semi-quantitative food-frequency questionnaire (FFQ) from 93 volunteers classified via colonoscopy examination according to the presence of intestinal polyps or adenocarcinoma. Cytokines, chemokines, and adipokines were determined from serum samples. We observed a reduction in adiponectin according to the damage to the mucosa, accompanied by an increase and decrease in C-X-C motif chemokine ligand 10 (CXCL10) and resistin, respectively, in CRC cases. The presence of aberrant crypt foci (ACF) in the polyp group was associated with higher tumor necrosis factor-alpha (TNF-α) concentrations. Vegetables were directly correlated with adiponectin and resistin levels, while the opposite occurred with red meat. A bioactive compound, soluble pectin, showed a negative association with TNF-α. Future dietary strategies could be developed to modulate specific immunological parameters in the context of CRC.

## 1. Introduction

According to GLOBOCAN data, about 1.9 million cases of colorectal cancer (CRC) occurred worldwide in 2020, this type of cancer ranking third in the global incidence of malignant tumors and second in mortality from cancer worldwide [1]. Despite improved survival and prognosis of CRC through primary prevention, the global number of new CRC diagnoses is expected to reach 3.2 million by 2040 [2]. The increasing incidence of this gastrointestinal pathology is mainly attributed to the exposure to different environmental risk factors resulting from lifestyle and dietary changes in Westernized countries [3]. In contrast to other cancers, in most cases, CRC precursor lesions develop slowly over years, if not decades, following the initial transformation of a normal colorectal epithelium to an adenoma [4]. Aberrant crypt foci (ACF) commonly appear as the first microscopic precursor lesions of the intestinal mucosa, being frequently concurrent with the presence of adenomas and hyperplastic polyps [5]. The interaction between systemic inflammation and the local immune response has been demonstrated to be one of the key factors involved in the initiation, development, and progression of several tumor types, including colon cancer [6,7]. Although the factors triggering the development of these lesions are not known exactly, some authors suggest that certain dietary compounds may modulate the colonic environment by modifying inflammation- and oxidative-stress-related parameters and promoting mutagenicity at the local level [8]. Among them, some antioxidants provided by vegetable foodstuffs and beverages exhibit anti-inflammatory actions, while red meat, ethanol, and fats have shown the opposite effect [9]. Closely linked to diet, overweight and obesity were related with a 10% increase in the risk of CRC in a population-based cohort study of 5.24 million U.K. adults [10]. Although the exact mechanisms connecting cancer and obesity are unknown, some of the proposed ways include the regulation of obesity-associated adipokines such as leptin, adiponectin, and resistin; an increase in plasma insulin and glucose intolerance; and other proinflammatory cytokines such as interleukin-6 (IL-6) and tumor necrosis factor-alpha (TNF-α) [11,12,13]. On one hand, it has been postulated that an increase in fat in the adipocytes can lead to increased blood levels of resistin, leptin, adiponectin, IL-6, and TNF-α, thereby enhancing the rate of growth, progression, and metastasis of tumors [13]. Results from in vitro studies also demonstrate that leptin activates the phosphoinositide 3-kinase (PI3K)/protein kinase B (PKB/AKT)/mammalian target of rapamycin (mTOR) signaling pathway, enhancing the proliferation of human colon cancer cells [14]. On the other hand, several authors pointed to a protective role of circulating adiponectin against CRC based on its anti-inflammatory and anti-neoplastic properties. This suggests a possible usefulness of this adipokine as a marker of the different stages of tumor progression [15,16,17,18,19,20]. Lifestyle factors such as physical activity and smoking were associated with adiponectin levels in a Japanese sample [21]. While more scientific evidence is needed, these data open new avenues of interest in the study of diet-mediated colon cancer prevention. Based on this background, the aim of this study was to describe the immunological and metabolic profile associated with intestinal mucosal lesions and to assess the potential impact of diet on these biomarkers.

## 2. Results

A general description of the study sample is presented in Table 1. The percentage of female volunteers in the control group was slightly higher, while male gender predominated in the other groups. The numbers of those playing sports and engaging in walking activity were statistically higher in the CRC group compared to the control group. In order to test whether these differences were attributable to gender, the data were examined by gender within each group, with no significant differences observed. Changes in the concentration of immunological parameters such as circulating adipokines, cytokines, and chemokines according to the clinical classification of the volunteers are represented in Figure 1 (Appendix A). The results revealed low levels of adiponectin and resistin in patients with CRC, with adiponectin also decreased in the presence of polyps. Conversely, the interferon-gamma (IFN-γ)-inducible C-X-C motif chemokine ligand 10 (CXCL10) was increased in the CRC group compared with controls, but no significant differences between the study groups were detected in the levels of the other chemokines or cytokines analyzed. Nevertheless, several molecules displayed an elevated dispersion, mainly in the polyp group.

The presence or absence of ACF in the group of volunteers diagnosed with intestinal polyps was examined to analyze whether, in the presence of ACF, there were differences in immunological parameters (Table 2). We observed statistically significantly increased levels of TNF-α. Chemokine (C-C motif) ligand 2 (CCL-2) also showed an increased concentration in the presence of ACF, near statistical significance (*p* = 0.067). Moreover, the TNF-α/Interleukin(IL)-10 ratio was also increased in patients with polyps and ACF (*p* = 0.035), thus supporting an inflammatory environment.

To explore the association in the sample between immunological parameters and the intake of food groups through diet, Spearman correlations were performed and heatmaps were plotted with the associations obtained (Figure 2). Among the immunological parameters, total cholesterol showed an inverse association with milk, dairy, and meat products, and more precisely with red meat. Red meat was also negatively correlated with resistin and IFN-γ. Both red and white meat were associated negatively with adiponectin, while vegetable intake showed the opposite tendency. Oils and fats were found to be directly correlated with the levels of leptin and inversely correlated with the levels of IFN-γ. Resistin showed a direct association with the consumption of fish, seafood, vegetables, and nuts and seeds, and an inverse association with the intake of cereal products, fruits, and red meat, as mentioned above. Alcoholic beverages were found to be positively correlated with IL-1β. The correlations of immunological parameters and food groups depending on the diagnosis groups were also evaluated (Appendix A). It was found that in the control and polyp groups, the negative correlation between red meat and adiponectin was maintained. The consumption of food groups was analyzed according to clinical diagnosis groups, showing a statistically significant increase in the consumption of alcoholic beverages in the polyp group with respect to controls (Appendix A). The CRC group did not show significant differences, possibly due to the low number of samples with available dietary information. When the differences in the intake of food groups were evaluated according to ACF presence in the polyp group, the subgroup diagnosed with ACF showed increased intake of seafood (Appendix A).

We looked deeper into the associations of immunological parameters and diet by analyzing the intake of their constituents, including macro- and micronutrients, bioactive compounds, and xenobiotic compounds (Figure 3). In concordance with the findings above, dietary proteins and lipids—more precisely, animal proteins and saturated, monounsaturated, and polyunsaturated fatty acids—were associated with decreased levels of adiponectin. The direction of this adiponectin association with dietary components was also maintained for vitamin B_2_, while it was inverted with soluble pectin. When evaluating leptin, monounsaturated fatty acids from oils and fats and minerals such as iron, magnesium, selenium, and sodium showed positive correlations with this immunological parameter.

Negative correlations were found with the xenobiotic 2-amino-3,4 dimethylimidazo (4,5,f) quinoline (MeIQ). Higher levels of resistin were associated with lower intakes of flavonoids and stilbenes. Soluble pectin displayed a similar negative association with both TNF-α and chemokine CCL2, interestingly similar to what was found in the polyp group with ACF presence. Lipids were the unique compounds inversely associated with IFN-γ. We found that IL-10 was negatively correlated with total protein and animal protein; with lipids; with vitamins such as vitamin A, B12, or B6; with several minerals; and with the xenobiotic benzo(a)pyrene (B(a)P), a polycyclic aromatic hydrocarbon (PAH), the latter also being negatively correlated with cholesterol levels. The correlations of immunological parameters and dietary compounds depending on the diagnosis groups were also evaluated (Appendix A). The intake of all these dietary compounds was evaluated according to clinical diagnosis groups, revealing a statistically significant increase in the intake of ethanol and dibenzo (a) anthracene (DiB(a)A) in the group of volunteers diagnosed with intestinal polyps compared to the control group (Appendix A). Moreover, we found higher concentrations of vitamin A and calcium in the polyp group compared to the CRC group. The intake of dietary compounds was also evaluated according to ACF presence in the polyp group, not showing any significant difference (Appendix A).

## 3. Discussion

The present study provides further insight into the existing knowledge on the impact of diet on the immune system at different stages of CRC development. Our findings pointed to CXCL10, adiponectin, and resistin as potential markers involved in the progression of damage on the colorectal mucosa. The association of some dietary components with these immunological parameters may help in the design of dietary strategies aimed at the primary or secondary prevention of this pathology [22,23].

The decrease in serum adiponectin accompanied by increased levels of CXCL10 in subjects from the polyp and CRC groups as compared to volunteers from the control group is one of the main findings of the present study. Clinically, adiponectin is used as a biomarker in obesity-related diseases, and its level showed an inverse relationship with increasing central adiposity and type 2 diabetes [24,25]. There is strong evidence supporting the assertion that low levels of plasma adiponectin promote the suppression of AMP-activated protein kinase (AMPK) activity, activating the mTOR pathway, which is directly related to the proliferation of colorectal epithelial cells and colorectal carcinogenesis [26]. Our results are in accordance with previous ones showing a reduction in adiponectin in cancer models, although there is no consensus across the literature [16,17,27,28]. Previous studies showed that modulation of the adiponectin concentration is possible by adhering to Mediterranean dietary patterns [29]. In this regard, we found negative correlations of adiponectin with red and white meat. Interestingly, fiber supplementation seemed to be an effective therapeutic way of increasing adiponectin levels in blood [30]. Accordingly, a positive correlation of adiponectin with vegetables and soluble pectin was observed in this study. Moreover, in the studied sample, CXCL10 levels increased and resistin levels decreased in CRC compared to the control and polyp groups, respectively. CXCL10 is considered a pro-inflammatory chemokine associated with lipotoxicity, and its upregulation has been described in liver injury in murine models [31,32]. CXCL10 was proposed as mediator of CD4+ T cell trafficking, while in mice with high-fat diets, an increase in the expression of this cytokine occurs in muscle [33,34]. This chemokine can contribute to the heat in tumor areas, promoting its development [35]. Recently, CXCL10 has gained attention due to its role in the clinical outcome of patients infected with SARS-CoV-2 [36].

In our sample population, we did not find any significant correlation of CXCL10 with diet, although when the polyp group was evaluated, a negative correlation with oils and fats was found. Regarding resistin, several authors reported higher levels of this cytokine in different inflammation-related disorders such as atherosclerosis, chronic inflammatory bowel disease, chronic renal disease, systemic lupus erythematosus (SLE), or arthritis [37,38,39]. The expression of resistin seems to be increased in the presence of high concentrations of other pro-inflammatory cytokines [40]. However, in our sample, the levels of resistin were reduced in the CRC group, and we could not corroborate any association with the pro-inflammatory cytokines IL17, IL6, IL12, or TNF-α. Recently, the European Prospective Investigation into Cancer and Nutrition (EPIC) study found no association between pre-diagnostic circulating resistin concentrations and the risk of CRC, suggesting the role of resistin as a possible marker of CRC instead of a risk factor [41]. In the present sample, several factors such as higher resistin levels in those of male gender (31.04 vs. 19.59 ng/mL), increased walking activity, or greater age in the CRC group may contribute to explaining why this group, composed exclusively of males, showed lower resistin levels compared to the control group, in accordance with a previous study enrolling Greek students that revealed increased resistin concentrations in healthy females compared to males [42]. Surprisingly, we found direct and inverse correlations of resistin with vegetables and red meat, respectively, probably related to the decreased consumption of vegetables in the CRC group, which also displayed lower resistin levels, or to the presence of any confounding factor we may be neglecting.

Moreover, when immunological parameters were evaluated in volunteers diagnosed with intestinal polyps and examined as a function of the presence or absence of ACF, TNF-α showed a statistically significant increase in subjects with ACF. Factors such as age, body mass index (BMI), and diet seem to affect the ACF number, which has been proposed to be utilized as a marker for the presence of lesions in the intestine [5,43]. TNF-α is a known inducer of NF-κB activity, secreted during the early phase of acute and chronic inflammatory diseases [44]. It is known that chronic inflammation leads to immune tolerance, thus promoting tumor formation and development [45]. Therefore, the modulation of inflammation in individuals with ACF could be a target for colon cancer prevention. In this sense, it has been reported that dietary components regulated the expression of TNF-α and other inflammatory cytokines and attenuated the progression of ACF in rat models [46]. In this regard, we previously described that the presence of ACF was accompanied by increased fecal mutagenicity in fecal samples of volunteers diagnosed with intestinal polyps [47]. The existence of several TNF inhibitors (e.g., infliximab, adalimumab, etc.) in the market and the current ongoing clinical trials highlight the clinical importance of this molecule in the treatment of diseases such as systemic rheumatic disease and inflammatory bowel disease [48]. TNF-α may also be downregulated by high adiponectin levels, thereby reducing TNF-α-associated inflammation, as suggested by several authors [49]. Furthermore, we found that the TNF-α/IL-10 ratio was increased in the presence of ACF. IL-10 shows opposite effects compared to TNF-α in inflammation, being mainly considered as an anti-inflammatory cytokine [50]. Given the opposite roles and the tightly regulated relationship between IL-10 and TNF-α levels, we consider the TNF-α/IL-10 ratio to be a better biomarker than any of the individual cytokines. This ratio has been found to be increased in infectious diseases and associated with poor health outcomes [50,51,52]. However, upregulation of IL-10 by either pharmacological (e.g., TNF-α blockage) or dietary interventions could be a promising target for individuals at risk. Many of the immunological parameters measured in this study are evaluated routinely by physicians. The existence of commercial panels makes the analysis of several parameters easy and affordable, and the information provided helps to associate abnormalities with a disease and its progression.

It is also important to note that in the population under study, the consumption of alcoholic beverages was higher in both the polyp group and the CRC group, with the intake of ethanol being statistically increased in the polyp group with respect to controls, as previously reported in individuals of this sample population [47]. Some authors found a clear association between high consumption of alcoholic beverages and mortality by CRC [53,54,55,56]. It is known that alcohol intake can contribute to initiating carcinogenic processes by destroying folate when the microbiota transforms ethanol into acetaldehyde in the colon. Subsequently, folate deficiency would cause chromosomal deterioration, inadequate incorporation of uracil into DNA, and other anomalies in DNA precursors, initiating tumor processes [57]. In the CRC group, the intake of red and processed meats and the consumption of alcoholic beverages was lower than that in the polyp group, which may be attributed to the possible existence of gastric symptoms accompanied by diet self-moderation previous to the clinical diagnosis of the CRC. In terms of immunological and dietary parameters, in the present study, volunteers from the polyp group generally showed values between those for the control and CRC groups, as many variables followed a continuous trend across the different stages of development of the disease. Interventions focusing on the group of volunteers diagnosed with intestinal polyps could be a useful approach to avoid the development of CRC. Specifically, dietetic strategies to reduce the levels of certain inflammatory parameters could be of paramount interest. In this regard, higher adherence to Mediterranean dietary patterns seems to be the best protective factor against gastrointestinal pathologies, being capable of reducing inflammatory parameters related with CRC processes [58,59]. This diet, rich in fruits, legumes, vegetables, olive oil, herbs, and spices, involves high intake of fiber and of polyphenols such as apigenin, curcumin, epigallocatechin gallate, quercetin-rutine, and resveratrol [60,61]. Moreover, the intrinsic abundant intake of antioxidant molecules is associated with health-promoting properties [61]. The Mediterranean diet also contributes to modulating potentially toxic bile acids and the gut microbiota, frequently altered in CRC development [62]. Another potentially protective dietary habit could be related with a decrease in the consumption of dietary emulsifiers and additives associated with CRC risk [63]. Interestingly, nutraceuticals derived from Mediterranean diet products have gained attention and are being proposed as specific dietary components for personalized adjuvant therapies in the prevention of CRC, by either reducing inflammation or preserving a healthy microbiota in the intestine [58,64].

Among the limitations of the present study, results from the CRC group should be considered with caution due to the limited sample size. Surprisingly, the subjects in this group were more active than those in the polyp or control group. These findings may be influenced by the small sample size. It cannot be discarded that, as a consequence of the discomfort caused by the development of the disease, they may have attempted to improve their lifestyle. Moreover, the diverse molecular characteristics and pathways of the intestinal polyps should be considered when analyzing higher intragroup variability in those volunteers diagnosed with polyps.

## 4. Materials and Methods

### 4.1. Study Design and Volunteers

This transversal analysis is part of the broader project “Effect of Diet and exposure to XEnobiotics generated during food processing on the genotoxic/cytotoxic capacity of the intestinal Microbiota” (MIXED).

The recruitment of volunteers and collection of blood and tissue samples was carried out from October 2019 to December 2021 by the faculties of the Digestive Section of the Central University Hospital of Asturias (HUCA) and the Carmen and Severo Ochoa Hospital in Asturias, in the north of Spain. Volunteers were selected from among patients who came to the hospital for consultation due to clinical symptoms and among those included in the colon cancer screening program in our region. The exclusion criteria applied to subjects with age under 40 years or over 75 years, as well as those receiving omeprazole, antibiotics, corticoids, or non-steroidal anti-inflammatory drugs. Also, having specific cancer treatment at the time of the study or in the previous two months, previous surgery of the digestive system, autoimmunity, altered thyroid function, or history of diabetes or goiter were considered as exclusion criteria. Those individuals interested in participating were informed of the objectives of the study and signed an informed consent form. Prior to the preparation of the volunteers for colonoscopy, blood samples were taken by venipuncture at proposed intervals between 09.00 and 11.00 after an overnight fast. Blood samples were collected in separate tubes for serum and plasma, immediately kept on ice, and centrifuged at 1000× *g*, 15 °C. The resultant aliquots were immediately frozen at −80 °C until analyses. A biopsy of intestinal mucosa from volunteers diagnosed with intestinal polyps was extracted during colonoscopy for examination of ACF at the Pathology Department. In total, 93 participants were included in the study. Subjects were classified into three clinical groups according to their colonoscopy results: control (*n* = 37), polyp presence (*n* = 49), and CRC (*n* = 7).

This project was evaluated and approved by the Regional Ethics Committee of Clinical Research of Asturias (Ref. 163/19) and by the Committee on Bioethics of CSIC (Ref. 174/2020). The procedures were performed in accordance with the fundamental principles set out in the Declaration of Helsinki, the Oviedo Bioethics Convention, and the Council of Europe Convention on Human Rights and Biomedicine, as well as in Spanish legislation on bioethics. Directive 95/46/EC of the European Parliament and the Council of October 1995, on the protection of individuals regarding the processing of personal data, was strictly followed.

### 4.2. Nutritional Assessment

Dietary information was obtained from patients when they arrived for their colonoscopy results at a medical consultation by means of a personalized interview conducted by trained interviewers. Exceptionally, as a result of the pandemic and COVID-19 restrictions on visitors to hospitals in Spain, some of the surveys were conducted through online tools. For this purpose, a semi-quantitative food-frequency questionnaire (FFQ) was constructed with 155 items. In addition to food and culinary preparations, the specific type of food was recorded, as well as cooking methods and other related information, when necessary. Information relating to dietary assessment has been previously published [47]. The classification of the food into food groups was carried out according to the Centre for Higher Education in Nutrition and Dietetics (CESNID) criteria [65]. Food composition tables from CESNID and the United States Department of Agriculture (USDA) were used to transform food consumption into energy and macronutrient intake [65,66]. The phenolic content of the foods was extracted using Phenol Explorer 3.6, and the fiber content was taken from the tables by Marlett and Cheung [67,68]. Oxygen Radical Activity Capacity (ORAC) was calculated according to the article by Wu et al. [69]. During personalized interviews, sleeping hours and physical activity were recorded as the self-referred time per day for each in the last year, while information on smoking habits was obtained by asking about cigarette smoking throughout life. Basal metabolic rate (BMR) was calculated using the Harris and Benedict formula.

### 4.3. Anthropometrical Determinations

Height (m) and weight (kg) were taken via standardized protocols [70]. Body mass index was calculated using the formula weight/(height)^2^. Subjects were classified into normal weight (18.5–24.9 kg/m^2^), overweight (25.0–29.9 kg/m^2^), and obese (30.0 kg/m^2^), based on the Spanish Society for the Study of Obesity (SEEDO) criteria [71].

### 4.4. Measurement of Immunological Parameters

To quantify the total antioxidant capacity of serum samples, a spectrophotometric method based on the ferric reducing antioxidant power (FRAP method) was performed using a commercial kit (Total antioxidant capacity (T-AOC) Assay Kit; ELK Biotechnology, Denver, CO, USA). Total cholesterol was analyzed enzymatically via the cholesterol oxidase e-p-aminophenazone (CHOD-PAP) method using a commercial kit (Total-Cholesterol Assay Kit; ELK Biotechnology, Denver, CO, USA). Circulating levels of cytokines, chemokines, and adipokines were quantified in serum samples using two pre-defined bead-based multiplex assays, following the protocol provided by the manufacturer and using a FACSCanto II flow cytometer (BD Biosciences, Franklin Lakes, NJ, USA). Serum samples were maintained at −80 °C until determinations. IL-4, IL-2, IP-10 (CXCL10), IL-1β, TNF-α, CCL2 (MCP-1), IL-17A, IL-6, IL-10, IFN-γ, IL-12p70, and free active TFGβ1 levels were assessed using the *Human Essential Immune Response Panel* (LEGENDplex, BioLegend, San Diego, CA, USA). The detection limits were 0.97 pg/mL, 1.81 pg/mL, 1.28 pg/mL, 0.65 pg/mL, 0.88 pg/mL, 1.45 pg/mL, 2.02 pg/mL, 0.97 pg/mL, 0.77 pg/mL, 0.76 pg/mL, 0.77 pg/mL, and 3.10 pg/mL, respectively. For adiponectin, adipsin, leptin, and resistin determination, the *Human Metabolic Panel 1* (LEGENDplex, BioLegend, San Diego, CA, USA) was used, with detection limits of 41.4 pg/mL, 5.4 pg/mL, 1.6 pg/mL, and 1.4 pg/mL, respectively.

### 4.5. Statistical Analyses

The results were analyzed using IBM SPSS software version 25.0 (IBM SPSS, Inc., Chicago, IL, USA) and RStudio software version 1.4.3 (Posit Software, Boston, MA, USA). GraphPad Prism 9 (GraphPad Software, Boston, MA, USA) and RStudio software were used for graphical representations. Overall, categorical variables were summarized as the number and percentage, and continuous ones were summarized as the mean and standard deviation. Fisher tests and *t*-tests were performed for categorical and continuous variables regarding a general description of the sample, respectively (*p*-value < 0.05). The goodness of fit to a normal distribution was checked by means of the Kolmogorov–Smirnov test. As normality of the immunological and dietary variables was not achieved, Mann–Whitney U tests were performed to detect group differences (*p*-value < 0.05). Correlations were assessed using Spearman rank tests to explore the associations between immunological parameters and food groups and compounds. Heatmaps were generated using the “corrplot” R package version 0.92.

## 5. Conclusions

The variation in the levels of adipokines and chemokines with the progression of intestinal mucosal damage revealed the potential use of certain immune parameters as markers of the disease. Moreover, TNF-α concentrations were increased in the presence of ACF, supporting its pro-inflammatory role. The modulation of immunological parameters in patients by shifting to healthy dietary habits could be a non-invasive approach of great interest for future studies.

## Figures and Tables

**Figure 1 ijms-24-16451-f001:**
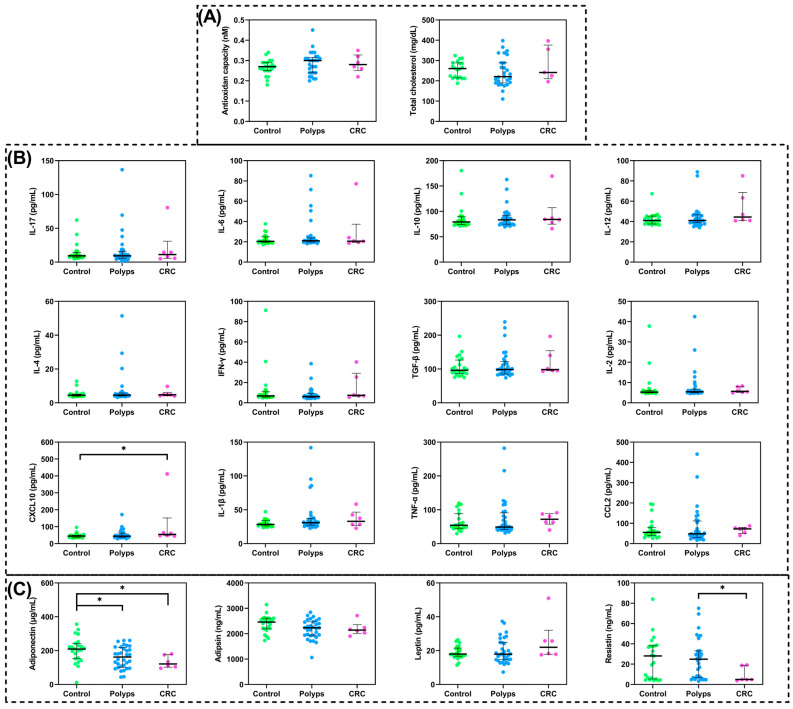
Differences in concentration of immunological parameters according to clinical diagnosis groups. (**A**) Antioxidant capacity and total cholesterol; (**B**) Cytokines and chemokines: IL-17, IL-6, IL-10, IL-12, IL-4, IFN-γ, TGF-β, IL-2, CXCL10, IL-1β, TNF-α, and CCL2; (**C**) Adipokines: Adiponectin, adipsin, leptin, and resistin. (*) Statistically significant differences between groups (*p* < 0.05) found via non-parametric tests. IL, interleukin; IFN-γ, interferon-gamma; TGF-β, transforming growth factor-beta; CXCL10, C-X-C motif chemokine ligand 10; TNF-α, tumor necrosis factor-alpha; CCL2, chemokine (C-C motif) ligand 2.

**Figure 2 ijms-24-16451-f002:**
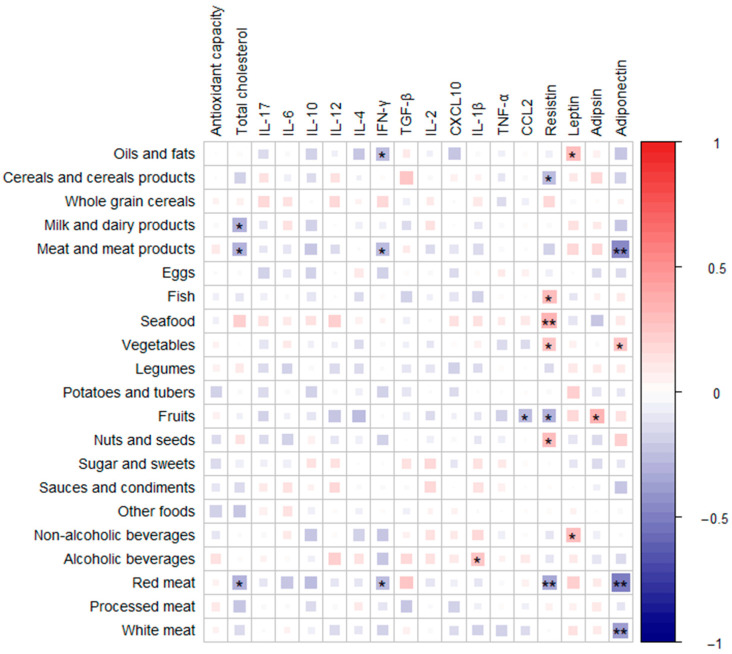
Heatmap defined by Spearman correlations between immunological parameters and food groups in the sample. Blue and red colors represent negative and positive associations, respectively. The color intensity is proportional to the degree of association. (*) *p* < 0.05, (**) *p* < 0.001. IL, interleukin; IFN-γ, interferon-gamma; TGF-β, transforming growth factor-beta; CXCL10, C-X-C motif chemokine ligand 10; TNF-α, tumor necrosis factor-alpha; CCL2, chemokine (C-C motif) ligand 2.

**Figure 3 ijms-24-16451-f003:**
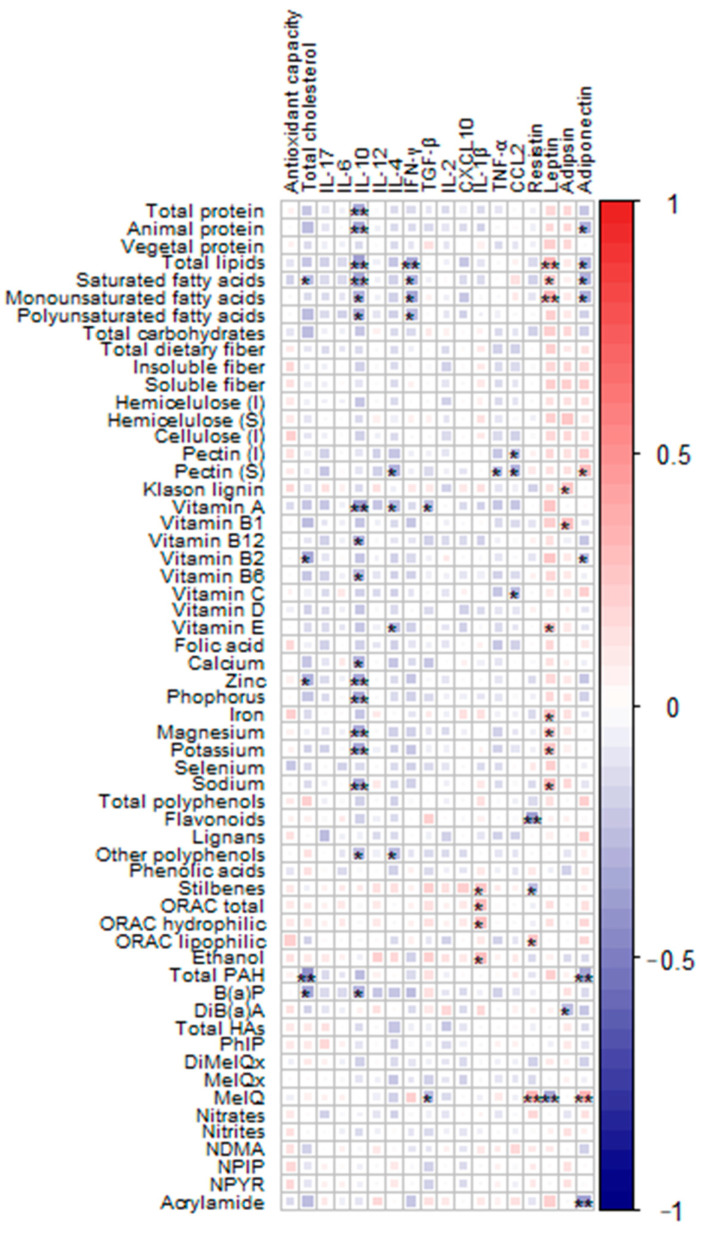
Heatmap defined by Spearman correlations between immunological parameters and dietary compounds in the sample. Blue and red colors represent negative and positive associations, respectively. The color intensity is proportional to the degree of association. (*) *p* < 0.05, (**) *p* < 0.001. IL, interleukin; IFN-γ, interferon-gamma; TGF-β, transforming growth factor-beta; CXCL10, C-X-C motif chemokine ligand 10; TNF-α, tumor necrosis factor-alpha; CCL2, chemokine (C-C motif) ligand 2; I, insoluble; S, soluble; ORAC, oxygen radical absorbance capacity; Total PAHs, total polycyclic aromatic hydrocarbons; B(a)P, benzo (a) pyrene; DiB(a)A, dibenzo (a) anthracene; Total HA, total heterocyclic amines; PhIP, 2-amino-1-methyl-6-phenylimidazo (4,5,b) pyridine; DiMeIQx, 2-amino-3,4,8 trimethylimidazo (4,5,f) quinoxaline; MeIQx, 2-amino-3,8 dimethylimidazo (4,5,f) quinoxaline; MeIQ, 2-amino-3,4 dimethylimidazo (4,5,f) quinoline; NDMA, N-nitrosodimethylamine; NPIP, N-nitrosopiperidine; NPYR, N-nitrosopyrrolidine.

**Table 1 ijms-24-16451-t001:** General description of the sample according to clinical diagnosis groups.

Variable		Control (*n* = 37)	Polyps (*n* = 49)	CRC (*n* = 7)
Gender	Male	17 (45.9%)	30 (61.2%)	7 (100.0%)
	Female	20 (54.1%)	19 (38.8%)	0 (0.0%)
Age (years)		59 ± 9	61 ± 6	64 ± 5
BMI (kg/m^2^)		26.17 ± 3.49	28.20 ± 4.41	26.25 ± 2.90
Energy intake (kcal/day)		2084.80 ± 759.64	2226.89 ± 848.08	2335.22 ± 540.20
Sport practice	Yes	9 (24.3%)	10 (20.4%)	4 (57.1%)
	No	28 (75.7%)	39 (79.6%)	3 (42.9%)
Sport activity (h/week)		1.04 ± 2.30	0.59 ± 1.43	4.29 ± 4.57 *
Walking activity (min/day)		57.77 ± 27.28	55.75 ± 29.52	83.57 ± 8.02 *
BMR (kcal/day)		1485.59 ± 230.60	1542.60 ± 253.91	1575.37 ± 126.01
Sleeping (h/day)		6.99 ± 0.92	6.82 ± 1.36	6.86 ± 0.90
Smoking habit	Current	6 (16.2%)	13 (26.5%)	1 (14.3%)
	Never	17 (45.9%)	21 (42.9%)	2 (28.6%)
	Former	14 (37.8%)	15 (30.6%)	4 (57.1%)

Values are shown as mean ± standard deviation (SD) for continuous variables or number (%) for categorical ones. (*) Statistically significant differences compared to control group (*p* < 0.05) found via *t*-test. BMI, body mass index; BMR, basal metabolic rate; CRC, colorectal cancer.

**Table 2 ijms-24-16451-t002:** Differences in immunological parameters according to aberrant crypt foci (ACF) presence in intestinal mucosa of volunteers diagnosed with intestinal polyps.

Parameter	ACF Presence	
	No	Yes
Antioxidant capacity (nM)	0.28 ± 0.05 (23)	0.33 ± 0.08 (6)
Total cholesterol (mg/dL)	230.20 ± 68.53 (22)	294.58 ± 71.28 (6)
Cytokines and chemokines		
IL-17 (pg/mL)	17.92 ± 28.90 (25)	16.43 ± 11.91 (6)
IL-6 (pg/mL)	26.85 ± 15.57 (25)	30.27 ± 20.32 (6)
IL-10 (pg/mL)	87.89 ± 21.61 (25)	89.35 ± 16.32 (6)
IL-12 (pg/mL)	43.06 ± 9.88 (25)	50.69 ± 18.99 (6)
IL-4 (pg/mL)	6.37 ± 5.86 (25)	12.55 ± 19.05 (6)
IFN-γ (pg/mL)	8.94 ± 7.56 (25)	7.13 ± 1.83 (6)
TGF-β (pg/mL)	109.35 ± 39.63 (25)	132.22 ± 42.51 (6)
IL-2 (pg/mL)	6.42 ± 2.55 (25)	15.54 ± 15.55 (6)
CXCL10 (pg/mL)	55.35 ± 31.44 (25)	51.93 ± 18.67 (6)
IL-1β (pg/mL)	37.22 ± 19.82 (25)	51.22 ± 44.89 (6)
TNF-α (pg/mL)	58.68 ± 27.47 (25)	131.24 ± 96.91 * (6)
CCL2 (pg/mL)	60.11 ± 47.50 (25)	171.69 ± 171.02 (6)
Adipokines		
Adiponectin (ng/mL)	163,146.13 ± 63,210.04 (24)	129,418.76 ± 57,864.85 (6)
Adipsin (ng/mL)	2153.22 ± 408.36 (24)	2284.82 ± 301.35 (6)
Leptin (pg/mL)	19.24 ± 7.70 (25)	21.27 ± 6.37 (6)
Resistin (ng/mL)	26.86 ± 20.71 (25)	23.72 ± 15.20 (6)

Values are shown as mean ± standard deviation (SD) and number of volunteers with available information in parentheses. (*) Statistically significant differences between groups (*p* < 0.05). Differences were analyzed via non-parametric tests. ACF, aberrant crypt foci; IL, interleukin; IFN-γ, interferon-gamma; TGF-β, transforming growth factor-beta; CXCL10, C-X-C motif chemokine ligand 10; TNF-α, tumor necrosis factor-alpha; CCL2, chemokine (C-C motif) ligand 2.

## Data Availability

Data are contained within the article and Appendix A.

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
