# Peer review of "Immunometabolic Profile Associated with Progressive Damage of the Intestinal Mucosa in Adults Screened for Colorectal Cancer: Association with Diet"

_ijms, 2023, doi:10.3390/ijms242216451_

Round 1
Reviewer 1 Report
Comments and Suggestions for Authors
In this paper, the authors explored the potential relationships between immunological biomarkers and diet in CRC-related lesions by obtaining dietary information and determining cytokines, chemokines and adipokines parameters from serum samples. They found that red meat, contrary to vegetables, was inversely correlated with adiponectin and resistin levels and pectin showed negative association with TNF-α. However, some issues need to be addressed.
1. According to our knowledge, physical activity and smoking should be able to affect our health, but in this paper, it seemed that physical activity and non-smoking had negative relations with CRCs, could the authors explain this.
2. The CRC group is so small population, which can affect the accuracy of the results.
3. According to the results, Il10 also can serve as a biomarker.
4. In Table S2 and S4, why there are only 3 samples for CRC group?
Comments on the Quality of English LanguageGood quality
Author Response
In this paper, the authors explored the potential relationships between immunological biomarkers and diet in CRC-related lesions by obtaining dietary information and determining cytokines, chemokines and adipokines parameters from serum samples. They found that red meat, contrary to vegetables, was inversely correlated with adiponectin and resistin levels and pectin showed negative association with TNF-α. However, some issues need to be addressed.
- According to our knowledge, physical activity and smoking should be able to affect our health, but in this paper, it seemed that physical activity and non-smoking had negative relations with CRCs, could the authors explain this.
We fully agree with the reviewer but given the observational nature of the work, we cannot establish directionality. Regarding smoking habit, although no significant differences were found in the prevalence of smokers between the groups analysed, the proportion is slightly higher in the polyp group. Therefore, the association does not appear to be positive “a priori”. We are more concerned about the differences observed with physical exercise, but we are unable to provide an explanation for this. We have introduced this aspect into the discussion (line 300, page 9): “results from CRC… improve their lifestyle”.
- The CRC group is so small population, which can affect the accuracy of the results.
Thank you for your comment. The control and polyp groups reach the sample size necessary to test the study hypothesis. We are aware that we cannot draw statistically robust conclusions in CRC group, but in our opinion it is interesting to include it because some of the observed results can serve as a basis for future studies.
- According to the results, Il10 also can serve as a biomarker.
This is an interesting point. Given the opposite roles and the tightly regulated relationship between IL-10 and TNFa levels, we consider TNFa/IL-10 ratio a better biomarker than any of the individual cytokines. However, upregulation of IL-10 by either pharmacological (e.g. TNFa blockage) or dietary interventions could be a promising target for individuals at risk. We have included it in the new version of the manuscript (page 9, lines 258-267).
- In Table S2 and S4, why there are only 3 samples for CRC group?
Among the CRC cases in which immunological parameters were measured, only 3 of them participated in dietary interviews. Therefore, dietary information is only available for three cases. We have commented this (line 138, page 5): “The CRC group … with available dietary information”
Reviewer 2 Report
Comments and Suggestions for Authors
This paper discusses the relationship between lifestyle, diet, and colorectal cancer (CRC) and its precursor lesions. The study aimed to explore the connections between immunological biomarkers and diet in individuals with CRC-related lesions. Dietary information was collected from 93 volunteers through a food-frequency questionnaire, and colonoscopy examinations categorized them based on the presence of intestinal polyps, adenocarcinoma, or no lesions. The study found that CRC and polyps were associated with alterations in certain immunological parameters, including decreased adiponectin levels and increased CXCL10 in CRC. Comparing polyps to CRC, there was a decrease in resistin in the CRC group, and the presence of aberrant crypt foci in the polyps group was linked to higher TNF-α concentrations. Additionally, red meat was inversely correlated with adiponectin and resistin levels, while soluble pectin showed a negative association with TNF-α. The findings suggest the potential for developing dietary strategies to modulate immunological parameters in the context of CRC. The topic is of interest and the paper is well written. The title is clear and concise, anyway, I recommend to write the abstract in a more attractive way, to involve readers to entry in the full text paper. From a methodological point of view, I am wondering if the low number of CRC patients (7) could affect the general interpretations of results. To this address I strongly recommend to calculate and report the statistical power for this study. The discussion section is well organized, but I would suggest to improve it from a clinical point view.
Comments on the Quality of English LanguageMinor editing of English language required
Author Response
This paper discusses the relationship between lifestyle, diet, and colorectal cancer (CRC) and its precursor lesions. The study aimed to explore the connections between immunological biomarkers and diet in individuals with CRC-related lesions. Dietary information was collected from 93 volunteers through a food-frequency questionnaire, and colonoscopy examinations categorized them based on the presence of intestinal polyps, adenocarcinoma, or no lesions. The study found that CRC and polyps were associated with alterations in certain immunological parameters, including decreased adiponectin levels and increased CXCL10 in CRC. Comparing polyps to CRC, there was a decrease in resistin in the CRC group, and the presence of aberrant crypt foci in the polyps group was linked to higher TNF-α concentrations. Additionally, red meat was inversely correlated with adiponectin and resistin levels, while soluble pectin showed a negative association with TNF-α. The findings suggest the potential for developing dietary strategies to modulate immunological parameters in the context of CRC. The topic is of interest and the paper is well written. The title is clear and concise, anyway, I recommend to write the abstract in a more attractive way, to involve readers to entry in the full text paper.
Thank you for your suggestion. The abstract has been rewritten to attract more attention from the potential readers.
From a methodological point of view, I am wondering if the low number of CRC patients (7) could affect the general interpretations of results. To this address I strongly recommend to calculate and report the statistical power for this study.
The statistical power for the comparisons in the study has been calculated using “pwr” package on R Studio. The control and polyp groups are of the necessary size for the statistical analyses performed. However, the CRC group does not have the minimum size necessary to be able to accept the null hypothesis with a 0.05 error. For this reason, this is reflected in the limitations section at the end of the discussion section.
The discussion section is well organized, but I would suggest to improve it from a clinical point view.
According to your comment more information regarding clinical features has been added in the discussion section:
Line 196, page 7: “Clinically, adiponectin is .. and type 2 diabetes”.
Line 214, page 7: “This chemokine can… infected with SARS-CoV-2”.
Line 251, page 8: “The existence of several… and inflammatory bowel disease”.
Line 263, page 8: “Many of the immunological… a disease and its progression”.
Reviewer 3 Report
Comments and Suggestions for Authors
In this work the authors aimed to study the potential relationships between the diet and immunological biomarkers in CRC-related lesions and CRC. I think this is a very interesting study and could prove very beneficial for the CRC patient. However, studies based on the patient’s memory and sincerity of their responses during the interview are always difficult to perform and to assign significance.
There are couple of concerns about this study that I state here point by point :
1. My first concern is the number of patients. The number of volunteers in this study is acceptable for control and polyps groups but CRC group has only 7patients and they are all male. This number is very small. Small number of CRC patients and also having only male patients might deviate the results.
2. Was the nutritional assessment protocol performed in the same interview? What period of time was taken into account? The evaluated period should be relatively long but there is a possibility of people not responding correctly, especially if they need to rely on their memory. Or did the authors ask about their usual foods without taking into account the real consumption?
3. Physical activity was measured as walking activity per day in the last years, declared by the patient. Was this the only physical activity performed by the patients? Some types of physical activity (like running, swimming or weightlifting) could deviate the results.
4. In the discussion section the authors should provide a small conclusion highlighting their most important findings and their implications. Especially because they finish this section with the study limitations and summarizing their findings could help put emphasis on their best results.
Comments on the Quality of English LanguageThere are some language errors throughout the manuscript and should be corrected by the native English speaker.
Author Response
In this work the authors aimed to study the potential relationships between the diet and immunological biomarkers in CRC-related lesions and CRC. I think this is a very interesting study and could prove very beneficial for the CRC patient. However, studies based on the patient’s memory and sincerity of their responses during the interview are always difficult to perform and to assign significance.
There are couple of concerns about this study that I state here point by point:
- My first concern is the number of patients. The number of volunteers in this study is acceptable for control and polyps groups but CRC group has only 7patients and they are all male. This number is very small. Small number of CRC patients and also having only male patients might deviate the results.
When performing statistical analysis for this study, we kept in mind that CRC group was composed of by few patients and only males, compared to the other groups of study. However, we though that the results obtained were enough interesting to include them in the work, at least, as preliminary or pilot results that may be later confirmed in other studies showing more statistical power.
- Was the nutritional assessment protocol performed in the same interview? What period of time was taken into account? The evaluated period should be relatively long but there is a possibility of people not responding correctly, especially if they need to rely on their memory. Or did the authors ask about their usual foods without taking into account the real consumption?
Due to SARS-COVID19 emergency situation some of the dietary interviews had to be performed using an online tool after patients went to collect their medical results. In this case, patients were asked to refer their height and weight based in their last medical report. In the interview, patients were asked about their consumption habits of the last year, being the minimum consumption frequency taken into account 1 month (1 time consumed per month). This methodology was particularly focused on collecting information about the consumption and cooking habits and preferences. We validated the developed FFQ comparing results with the considered gold standard for the measurement of food amount consumption: A 24h dietary recall. It was performed in the following paper:
Zapico, A.; Ruiz-Saavedra, S.; Gómez-Martín, M.; de los Reyes-Gavilán, C.G.; González, S. Pilot Study for the Dietary Assessment of Xenobiotics Derived from Food Processing in an Adult Spanish Sample. Foods 2022, 11, 470. https://doi.org/10.3390/foods11030470
- Physical activity was measured as walking activity per day in the last years, declared by the patient. Was this the only physical activity performed by the patients? Some types of physical activity (like running, swimming or weightlifting) could deviate the results.
We decided to include this information because in most cases, patients did not report other physical activity apart of walking. However, to add more information in this regard, we have included in Table 1 variables such as sport practice, time per day of sport and basal metabolic rate (BMR). This information is now available in line 364, page 11: “sleeping hours and physical activity”.
Moreover, we have performed some analyses not shown in the manuscript comparing variables according to gender, without detecting any statistical significant different:
Male |
Female |
|
Sport activity (h/week) |
1.41 ± 2.85 |
0.55 ± 1.15 |
Walking activity (min/day) |
58.72 ± 30.83 |
58.63 ± 24.87 |
MBR |
242.93 ± 135.62 |
212.37 ± 97.01 |
Control |
Polyps |
CRC |
|||
Male |
Female |
Male |
Female |
Male |
|
Sport activity (h/week) |
1.59 ± 3.02 |
0.58 ± 1.37 |
0.63 ± 1.69 |
0.53 ± 0.9 |
4.29 ± 4.57 |
Walking activity (min/day) |
53.65 ± 30.28 |
61.28 ± 24.7 |
55.69 ± 32.37 |
55.84 ± 25.42 |
83.57 ± 8.02 |
MBR |
231.49 ± 90.25 |
221.92 ± 98.28 |
206.24 ± 125.05 |
202.32 ± 97.28 |
422.72 ± 141.78 |
- In the discussion section the authors should provide a small conclusion highlighting their most important findings and their implications. Especially because they finish this section with the study limitations and summarizing their findings could help put emphasis on their best results.
Thank you for your comment. We have added a conclusions section to improve this. Also the text has been revised and edited.
Round 2
Reviewer 2 Report
Comments and Suggestions for Authors
The paper has been improved with the proposed suggestions. Thank you.
Reviewer 3 Report
Comments and Suggestions for Authors
I thank the authors for their efforts to address all my concerns. I consider the manuscript sufficiently improved and ready to be published.
Comments on the Quality of English LanguageMinor language editing is necessary